# Biodegradable Stent with mTOR Inhibitor-Eluting Reduces Progression of Ureteral Stricture

**DOI:** 10.3390/ijms22115664

**Published:** 2021-05-26

**Authors:** Dong-Ru Ho, Shih-Horng Su, Pey-Jium Chang, Wei-Yu Lin, Yun-Ching Huang, Jian-Hui Lin, Kuo-Tsai Huang, Wai-Nga Chan, Chih-Shou Chen

**Affiliations:** 1Division of Urology, Department of Surgery, Chang Gung Memorial Hospital, Chiayi 613016, Taiwan; redoxdrh@gmail.com (D.-R.H.); checotrade@gmail.com (W.-Y.L.); abradva@gmail.com (Y.-C.H.); mdrh20@ieee.org (J.-H.L.); ronsolglobalinc@gmail.com (K.-T.H.); yogochencjh@gmail.com (W.-N.C.); 2Department of Medicine, College of Medicine, Chang Gung University, Taoyuan 333323, Taiwan; r91a21015@mail.ntu.edu.tw; 3Department of Nursing, Chang Gung University of Science and Technology, Chiayi 613016, Taiwan; 4DuNing Incorperated, Tustin, CA 92780, USA; susolomon@gmail.com

**Keywords:** ureter, drug-eluting stent, biodegradable, ureteral stricture, rapamycin, sirolimus, mammalian target of rapamycin, fibrosis, epithelial–mesenchymal transition

## Abstract

In this study, we investigated the effect of mTOR inhibitor (mTORi) drug-eluting biodegradable stent (DE stent), a putative restenosis-inhibiting device for coronary artery, on thermal-injury-related ureteral stricture in rabbits. In vitro evaluation confirmed the dose-dependent effect of mTORi, i.e., rapamycin, on fibrotic markers in ureteral component cell lines. Upper ureteral fibrosis was induced by ureteral thermal injury in open surgery, which was followed by insertion of biodegradable stents, with or without rapamycin drug-eluting. Immunohistochemistry and Western blotting were performed 4 weeks after the operation to determine gross anatomy changes, collagen deposition, expression of epithelial–mesenchymal transition markers, including Smad, α-SMA, and SNAI 1. Ureteral thermal injury resulted in severe ipsilateral hydronephrosis. The levels of type III collagen, Smad, α-SMA, and SNAI 1 were increased 28 days after ureteral thermal injury. Treatment with mTORi-eluting biodegradable stents significantly attenuated thermal injury-induced urinary tract obstruction and reduced the level of fibrosis proteins, i.e., type III collagen. TGF-β and EMT signaling pathway markers, Smad and SNAI 1, were significantly modified in DE stent-treated thermal-injury-related ureteral stricture rabbits. These results suggested that intra-ureteral administration of rapamycin by DE stent provides modification of fibrosis signaling pathway, and inhibiting mTOR may result in fibrotic process change.

## 1. Introduction

Ureteral stricture is defined as ureter narrowing causing functional obstruction. It often results from aging, stones, surgical injury, malignancy, radiotherapy, inflammation, fibrosis, infection, ischemia, and trauma [1,2,3,4]. In particular, the well-accepted flexible ureteroscopy for upper tract stone management has led to increased incidence of iatrogenic ureteral stricture [5]. Early ureteroscopy studies reported ureteral stricture rates around 3–11%. The rate in recent studies decreased to less than 2% due to the advance of flexible ureteroscope, laser, access sheath, and minor traumatic instruments [6]. Ureter injuries can be well managed by experienced urological surgeons [7]. Moreover, different kinds of ureter stricture and stenosis can be managed with pyeloplasty [8], ureteroureterostomy/transureteroureterostomy [9,10], or psoas hitch/Boari flap [11] in upper, middle, or lower ureter injuries, respectively. However, fibrosis is always a challenge for surgeons because this abnormal healing process may ruin a successful operation. Furthermore, ureteral damage due to an energy-based surgical device has gradually increased. In this study, we would use electrocoagulation to reproduce thermal injury that resulted in rabbit ureteral stricture.

Ureteral strictures can be managed by endoscopic or balloon dilatation [12] or laparoscopic or open surgical ureteroplasty [13,14]. Ureteroplasty with buccal mucosal graft [15], ileal interposition, appendiceal onlay flap [16], etc. has been reported as effective ways to deal with recurrent ureter stricture. After the operation, ureteral stents are often inserted to secure ureteral patency and improve morbidity [17]. Ureteral stricture recurs if it is the result of a persistent fibrosis process. Ueshima et al. reported that transforming growth factor (TGF)-beta 1 mediated scar formation is considered to underlie recurrent ureter stricture [18]. Alpha-smooth muscle actin (α-SMA) expression by fibroblasts and collagen deposition have been found in scarring of the ureter and are associated with increased TGF-β expression in both fibroblasts and macrophages. TGF-β is the key factor of fibrosis, downstream collagen deposition, and epithelial–mesenchymal transition cascade. It can further activate mammalian target of rapamycin complex-1 to promote collagen production by increasing HIF-1 alpha expression [19].

Mammalian Target Of Rapamycin (mTOR) is a serine/theronine kinase that plays an important role in regulating collagen expression, and its inhibition effect induces a decreased collagen deposition in ECM [20,21,22,23]. By knocking down mTOR, a 70% reduction in fibroblast proliferation and simultaneous down-regulation in the expression of type I collagen in fibroblasts were observed [24]. It may also affect proliferating cell nuclear antigen, cyclin D1, collagen, fibronectin and α-SMA expression in fibroblasts [25]. Since TGF-β contributes to fibrosis through the mTOR mechanism, it is hypothesized that the local release of mTOR inhibitors (mTORi) contributes to the improvement of ureteral stricture.

Drug-eluting stents are a unique way to locally treat ureter diseases. They have been used for preventing infection, treating cancer, and killing pain [26,27,28,29,30]. However, such stents are either temporary or permanent implants with main complications such as dislocation, infection, pain, and encrustation. Additional post-stenting surgical and pharmaceutical cares frequently increase the workload of urologists. The advance of bioresorbable stent has partially addressed those complications [31,32,33]. Furthermore, biodegradable stents can be loaded with drugs using supercritical fluid CO2-impregnating techniques, which makes it a potentially effective and sustainable drug delivery system within the urinary tract [34,35,36,37]. The biodegradable stent is thus considered a promising device for local treatment of ureter diseases. Moreover, the technology of rapamycin-eluting bioresorbable stent has become mature in treating coronary artery narrowing in the past decade [38,39]. In this study, a rapamycin-eluting bioresorbable stent is employed to test the hypothesis of improving ureteral stricture with local release of mTOR inhibitors (mTORi).

In this feasibility study, we evaluate the expression profiles of TGF-β-related signaling as well as collagen deposition in ureter cross-section using a rabbit model of thermally injured ureteral stricture. To further elucidate the role of biodegradable stent and rapamycin in ureteral fibrosis, we investigated, using this same rabbit model of ureteral stricture, the effectiveness of rapamycin drug-eluting biodegradable stent on ureteral stricture reduction.

## 2. Results

### 2.1. In Vitro Dose-Dependent Effect of mTORi on Fibroblast and Urothelial Cell Line

In a urothelial cell line, α-SMA expression over b-actin decreased significantly when rapamycin up to 0.1nM was added (0.97 + 0.06), as 41% decreased compared with control (1.66 + 0.39, double asterisk, Figure 1A). In fibroblast, α-SMA expression decreased with rapamycin (0.53 + 0.08), a 37% decrease compared with control (0.84 + 0.23, double asterisk, Figure 1B). However, TGF-β cotreatment promoted fibroblast expression of α-SMA and activated Smad 2/3 significantly (* and #, Figure 1B, and hashtag, Figure 1D). In the urothelial cell line, on the contrary, a decrease in α-SMA and Smad 2/3 after treatment with TGF-β (* and #, Figure 1A, and * and #, Figure 1C) was found. Dose-dependent effects of mTOR inhibition on Smad phosphorylation and expression were both confirmed in fibroblast and urothelial cell line (double plus, Figure 1C, and double plus, Figure 1D).

Rapamycin (range, 0–100 nM) had no significant cytotoxic effects on TGF-β-induced fibroblast, but it significantly reduced the expression levels of α-SMA in a dose-dependent manner [19]. Smad phosphorylation and expression were also decreased after mTOR inhibition, both in fibroblast and urothelial cell lines. Previous studies have proven that TGF-β stimulation of mTOR complex requires Smad. Our study demonstrated that rapamycin significantly reduced the expression level of Smad in TGF-β-induced fibroblast in a dose-dependent manner. For the urothelial cell line, Smad expression was less promoted by TGF-β.

### 2.2. In Vivo Rabbit Segmental Ureteral Thermal Injury Model

#### 2.2.1. Schematic Procedure and Gross Anatomy Changes in Upper Urinary Tract of Animal Model

The stents were inserted as in Figure 2A after bilateral retroperitoneal dissection. Classification of groups includes sham operation control, thermal injury with DE stent, and thermal injury with NDE stent. The kidneys and upper ureters dissected from these groups are shown in Figure 2C. Rapamycin drug-eluting stents (DEs) or non-drug-eluting stents (NDEs) were inserted at thermally injured ureter proximal and distal to ureter incision, respectively (Figure 2B). One centimeter-long ureteral thermally injured segment was created. Stents were then introduced and deflated in these segments. After 4 weeks of healing, gross anatomy changes of the kidneys in Figure 2C were shown in Figure 2D.

#### 2.2.2. DE Stent Alleviate Fibrosis in Histopathology Examination

Loss of smooth muscle in para-ureteral region is prominent in thermally injured ureters (B, C, E, F, H, I). The circular smooth muscle (arrowhead, Figure 3A) decreased when ureter fibrosis dominated (asterisk, Figure 3B). Ureter integrity is moderately compromised in thermally injured ureters compared with control. Smooth muscle around ureter (arrowhead, Figure 3A) actually disappeared in thermally injured ureters (Figure 3B,C), along with hyperplasia of urothelium and increased thickness of periurethral soft tissue. Suburothelial collagen deposition is significant (Figure 3B,E). Collagen deposition is also prominent (hashtag, Figure 3H), as demonstrated by trichrome stain. When drug-eluting stent was applied, collagen deposition decreased (plus, Figure 3I, decreased blue stain in suburothelial region).

#### 2.2.3. Thermal Injury Promotes Type I and III Collagen Expression in Ureter 4 Weeks after Treatment

The major extracellular proteins expressed in thermally injured ureter segments are type I and III collagen. Thermal injury resulted in significant COL3A1 (*, Figure 4A) and COL1A2 (*, Figure 4B) expression in ureter segments. The amount of collagen deposition increased 89.2 ± 24.4% and 18.1 ± 14.8% in type III and I collagen, respectively. When thermally injured ureter segments are stented, the collagen expression was not significantly different between NDE and DE stent groups.

#### 2.2.4. Eluted mTOR Inhibitor Downregulate Smooth Muscle Actin and Upregulate Epithelial-Mesenchymal Transition

Collagen COL3A1 expression increased in thermally injured ureters (*, Figure 5B). Rapamycin eluted from stent helped decrease the amount of collagen deposition (*, Figure 5C). α-SMA expression decreased significantly due to loss of smooth muscle and myofibroblast in injured ureters (arrowhead, Figure 5E,F). Smad decreased when the ureter was exposed to rapamycin (Figure 5H,I and Figure 6B), but phosphorylated Smad actually increased significantly (*, Figure 6B). For EMT signaling, prominent Vimentin increased when the ureter was injured (Figure 5K) but decreased when exposed to rapamycin (Figure 5L). E-cadherin expression is mainly localized in the urothelium. It decreased even further in injured ureters (Figure 5N,O).

#### 2.2.5. Semi-Quantitative Analysis

The abundant α-SMA expression over GAPDH was found in control ureters (*, Figure 6A). Its expression is even less in drug-eluting stent compared with its non-eluting counterpart (double *, Figure 6A). This demonstrates more prominent α-SMA in control ureters compare to stented ureters, which is compatible with loss of smooth muscle in injured ureters (arrowhead, Figure 3A). Total Smad decreased in thermally injured ureters despite rapamycin treatment. Phosphorylated Smad (p-Smad) increased in thermally injured ureters, while total Smad decreased. This resulted in a significant change in active Smad ratio in rapamycin-treated ureter (*, Figure 6B). For EMT signaling, Vimentin expression increased in thermally injured ureters but was not significantly reduced under rapamycin treatment (Figure 6C). SNAI1 was mildly decreased in thermally injured ureters and significantly less compared to control when treated with rapamycin eluted stent (Figure 6D).

## 3. Discussion

The most significant finding of this study is that drug-eluting biodegradable stent can modify the ureteral reconstruction process. Important pathways, including TGF-β and EMT, could be modified through local eluted rapamycin. The decreased protein expressions of vimentin, SNAI1, and Smad are the most significant finding in signaling pathway changes related to rapamycin [40]. Though not significant, the result of remodeling is also affected. Collagen deposition decreased in the drug-eluting stent group compared with the non-drug eluting counterpart.

What was not expected is that ureter structural change occurred after thermal injury regardless of stent type. All ureters with thermal injury demonstrated a dilated lumen and paraurethral soft tissue accumulation. These could not be reversed by rapamycin drug-eluting stent. Although gross kidney size changes were minimal, dilated renal pelvis is still obvious (Figure 2D). Gross ureter change was prominent with the accumulation of suburothelium soft tissue compared with sham control (Figure 2C).

Rapamycin-eluting stent represents a breakthrough technology that has profoundly impacted the treatment of coronary artery disease [41]. There are different concerns when applying biodegradable stents in coronary arteries and ureters. For cardiac stents, biodegradable polymer (lactic and glycolic acid) was used for drug-eluting, while stainless steel platforms provide persist vessel dilation [42]. For ureters, biodegradable stents are used for preventing complications like infection and misplacement. Moreover, since there are few thrombotic events when a stent is used in the ureter, drug-eluting stents have been mainly used for cancer and pain of the upper urinary tract in the past [34,35].

We took the study of Anidjar et al. as a reference and used the same thermal radiofrequency electrocautery energy source [43]. However, we tried it in rabbits. The majority of ureter stent studies used pigs as models. These studies tested different surgical procedures and types of ureteral stents [44,45,46,47]. Rabbits [48,49] and canines [32] have been used for pyeloplasty and inflammation within urinary tract reconstruction. Since we were checking ureter remodeling after its injury, and well-established drug-eluting cardiac stents are around 2.5–10 mm in diameter, which suits rabbit ureter well, other energy sources including electroporation seem promising [50], which may be taken into account in further studies.

The benefit of our model is that control and drug-eluting stents were inserted in the same ureter. The mortality of rabbits was lower because the other sham-operated ureter can always maintain basic renal function for the animal. The drug eluted from the stent may not affect the proximal ureter due to urine flow direction. The shortcoming of our model is that local circulation of the proximal and distal ureter may be different. However, since circulation between bilateral ureters may also be different, keeping the thermally injured ureter on the same side may be the better way for this pilot study.

Although unilateral ureteral obstruction (UUO) is a model initially set up in mice for the study renal fibrosis, including BALB/c [51,52], C57BL/6 [53,54], ICR [55,56], etc., rats have also been used for the same purpose [57]. Sprague–Dawley rats [58,59], along with Wistar rats [60,61], have been both used to study renal fibrosis in UUO. Even porcine models have been established [62,63]. Previous studies have focused on the effect of mesenchymal stem cells on rat ureteral stricture [64]. We modified the experiment but focused on using drug-eluting biodegradable stents to treat ureteral fibrotic processes, which is the primary cause of hydronephrosis and renal fibrosis.

In this animal study, fibrotic tissue occupied most suburothelium and little α-SMA expression was found in thermally injured ureters. According to Zhao et. al, α-SMA was mainly detected in smooth muscle cells lining intramuscular blood vessel walls, but not in collagen-expressing cells [65]. The level of α-SMA expression by intramuscular fibrogenic cells may not correlate positively with the level of collagen gene expression. After 4 weeks of wound healing, ureter fibrosis is dominant, and smooth muscle loss prevails. Furthermore, persistent release of rapamycin from stents may result in an even lower α-SMA expression due to inhibition of the mTOR pathway.

Rapamycin-eluting biodegradable stents have been used in coronary artery diseases and have been shown to inhibit vascular smooth muscle cell proliferation [66,67]. This may help prevent restenosis of the vessels due to increased cells at the injury site. Furthermore, myofibroblast growth has also been found to be inhibited by mTOR inhibitors in the kidney [68], liver [69], cornea [20], and nasal cavity [70]. Although cell growth and numbers were not quantitatively evaluated in our study, cell nuclei numbers can be found decreased in the * region between Figure 5B,C along with collagen deposition. This may imply that cell proliferation was affected. Further quantitative studies are necessary to reveal the effect of rapamycin-eluting stents on cell proliferation.

## 4. Materials and Methods

### 4.1. Cell Line

The SV-HUC-1 and NIH/3T3 cell lines were purchased from the American Type Culture Collection (ATCC, Manassas, VA, USA) and used for all experiments. Cells were cultured in Ham’s F-12 and DMEM (Sigma-Aldrich, St. Louis, MO, USA) medium, respectively. The medium was supplemented with 10% fetal bovine serum (Hyclone, Logan, UT, USA), 100 units/mL penicillin, and 100 μg/mL streptomycin at 37 °C with 5% CO_2_ [71]. SV-HUC-1 and NIH/3T3 cells were seeded in 60 mm plastic tissue culture dishes. When cell confluence reached 80%, SV-HUC-1 cells were treated with different concentrations of rapamycin (0, 0.01, 0.1, 1, 10, 100 mM) with or without TGF-β 10 ng/mL cotreatment

### 4.2. Drug-Eluting Biodegradable Stent

The Mirage sirolimus-eluting bioresorbable microfiber stents were selected for this study due to their small strut size, built-in fluid dynamic feature, and predictable biodegradation time. Stent material is polylactic acid with <5% of dextrorotary isomer [72]. It is constructed with circular monofilament (diameter 108 um) ply in a helix coil configuration mounted with 3 backbones and a biodegradable PLA abluminal coating, which contains rapamycin (9 µg/mm) as antiproliferative agent [73]. In this study, stents with lengths of 10 mm (model # MMSES 27510) were deployed with diameters between 2.5 to 3.0 mm.

### 4.3. Animal Study

All the experimental procedures were approved by the Chang Gung Memorial Hospital Institutional Animal Care and Use Committee (approval number 2017122202), and complied with Scientific Application of Animals Chapter of Taiwan Animal Protection Law. A total of 12 New Zealand White rabbits were obtained from a commercial breeder and maintained on commercial feed, which was available ad libitum with water until immediately prior to the procedure. One side of rabbits’ ureter was randomly chosen as sham operation control, while the other was used for thermal injury and stent insertion. The stents were inserted as shown in Figure 2. Classification of groups included sham operation control, thermal injury with no-drug stent, and thermal injury with DE stent.

Once the rabbit had been anesthetized with halothane, the retroperitoneal space was dissected with bilateral paraspinal dorsal incision. The kidneys and upper ureters were freed from adjacent tissue, and an incision wound was made at 2 cm inferior to the ureteropelvic junction. Guidewire was inserted to facilitate electrocautery and stent insertion. Following Anidjar’s protocol of 10 watt thermal electrocautery for 10 s [43], a 1 cm-long ureteral stricture segment was created. Stents were then introduced and deflated in the thermally injured segments. A rapamycin-eluting stent (DE) or non-drug-eluting stent (NS) was inserted at the thermally injured ureter proximal and distal to ureter incision respectively (Figure 2B). In the control group, the same incision was made for traditional non-degradable 3 French ureteral stent insertion but without the application of thermal energy. Animals were sacrificed 4 weeks by pentobarbital overdose (125 mg/kg) for tissue collection [43]. The kidney and ureter were then harvested for gross anatomy observation and Western Blot analysis.

### 4.4. Histological Analysis

Bilateral ureter segments and kidneys of all rabbits were excised 4 weeks after ureter dissection with/without thermal injury. The tissues were embedded in paraffin. Samples were then serially sectioned at 7 µm thickness, and microscopic examination was conducted in tissue sections stained with hematoxylin and eosin, Masson’s trichrome stain, and immunohistochemistry.

### 4.5. Western Blot Analysis

Ureter segments from thermal injury and control groups were surgically separated using scissors under the microscope for Western blot analysis. Equal amounts of proteins were separated on SDS-PAGE gels and transferred to polyvinylidene fluoride membranes, which were then incubated overnight at 4 °C with primary antibodies. All experiments were repeated on at least 3 separate occasions. Primary antibodies used were mouse anti-α-SMA (sc-32251) (1:250 (*v*/*v*)), Smad2/3 (sc-133098) and -COL1A2 (sc-393573) (both 1:125 (*v*/*v*). Rabbit anti-SNAI 1 (H-130, sc-28199), Vimentin (C-20, sc-7557-R), E-cadherin (H-108, sc-7870) (all 1:125 (*v*/*v*)), and anti-COL3A1 (sc-8780-R), COL4A2 (sc-70246) (all 1:250 (*v*/*v*)). Goat anti-GAPDH (V-18, sc-20357) and mouse anti-b-Actin (C4, sc-47778) (both 1:20,000 (*v*/*v*)) were purchased from Santa Cruz Biotechnology. Appropriate secondary horseradish peroxidase-linked antibodies (Amersham) or (Santa Cruz Biotechnology) were used at between 1:1000 (*v*/*v*) and 1:10,000 (*v*/*v*), incubated in SuperSignal™ West Femto Maximum Sensitivity Substrate (Thermo Fisher Scientific, Tewksbury, MA, USA), and then imaged on a G:BOX Chemi XX9 (Syngene, Frederick, MD, USA). As a loading control, membranes were stripped using Restore™ Western Blot Stripping Buffer (Thermo Fisher Scientific, Tewksbury, MA, USA) and re-probed with mouse anti-b-actin. The volume (arbitrary units and intensity) of each protein species was determined using Quantity One software (Bio-Rad). The average value of control sample data was used to calculate the relative expression ratio as fold changes in each dataset of ureteral thermal injury and control samples.

### 4.6. Statistics

All values were presented as the mean ± SEM. An unpaired t-test was used to test for differences between groups. All tests were two-sided, and *p* values < 0.05 were considered statistically significant. All statistical analyses were performed using PRISM (GraphPad Software, La Jolla, CA, USA).

## 5. Conclusions

Because administration of rapamycin drug-eluting biodegradable stent effectively improved collagen deposition after ureteral thermal injury, application of drug-eluting stents could be an important way to reduce ureter stricture, and blockade of mTOR in the ureter remodeling process has therapeutic potential for ureteral stricture attributable to iatrogenic injuries.

## Figures and Tables

**Figure 1 ijms-22-05664-f001:**
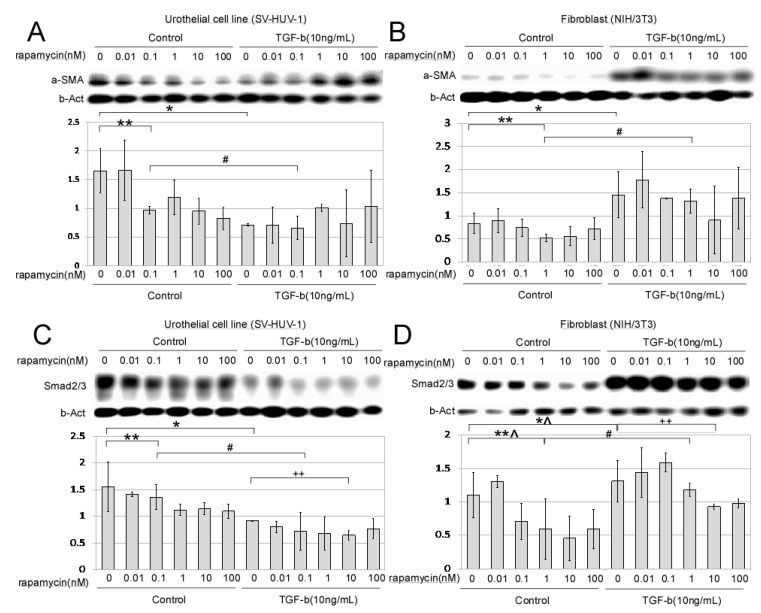
The dose-dependent effect of rapamycin on alpha-smooth muscle actin and Smad protein expression in fibroblast and urothelial cell line. α-SMA expression in (**A**) urothelial cell line and (**B**) fibroblast cell line with or without TGF-b treatment, demonstrating dose-dependent effect of rapamycin. Smad 2/3 expression in (**C**) urothelial cell line and (**D**) fibroblast cell line with or without TGF-b treatment, demonstrating dose-dependent effect of rapamycin. * *p* < 0.05 represents a significant difference for the TGF-b group compared with the control without TGF-b group, both under no rapamycin treatment. ** *p* < 0.05 represents a significant difference for the rapamycin treated group compared with the control without rapamycin group, both under no TGF-b treatment. # *p* < 0.05 represents a significant difference for the TGF-b group compared with the control without TGF-b group, both under specific rapamycin treatment concentration, i.e., 1 nM for fibroblast and 0.1 nM for urothelial cell. *^ *p* > 0.05 represents an insignificant difference for the TGF-b group compared with the control without TGF-b group, both under no rapamycin treatment. **^ *p* > 0.05 represents an insignificant difference for the rapamycin treated group compared with the control without rapamycin group, both under no TGF-b treatment. ++ *p* < 0.05 represents a significant difference for the rapamycin treated group compared with the control without rapamycin group, both under 10 ng/dL TGF-b treatment.

**Figure 2 ijms-22-05664-f002:**
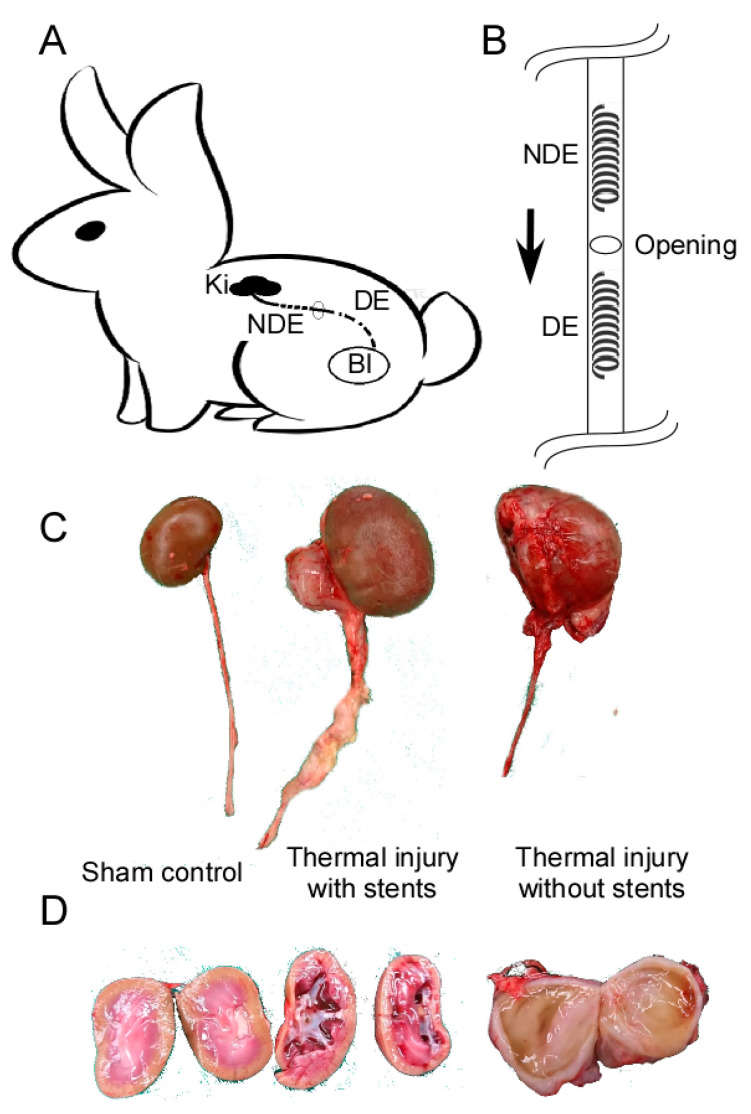
Operation method (**A**). Retroperitoneal space in the paraspinal region was dissected. Ureteral stents were inserted through the mid-ureteral opening. (**B**). In thermal injury with the stents group, a non-drug-eluting stent (NDE) was inserted in the upper ureter near the kidney (Ki), while a drug-eluting stent (DE) was inserted in the distal ureter near the bladder (Bl). The eluted drug may affect the distal ureter segment more due to urine flow direction (arrow). (**C**). Gross appearance of the kidney and ureter after 4 weeks of wound healing in sham control, thermal injury with stents, and thermal injury without stents. (**D**). Collecting system appearance in sham control, thermal injury with stents, and thermal injury without stents groups. The mild dilated renal pelvis can still be noted in the collecting system of the stented ureter. Prominent hydronephrosis if no stent was inserted after thermal injury.

**Figure 3 ijms-22-05664-f003:**
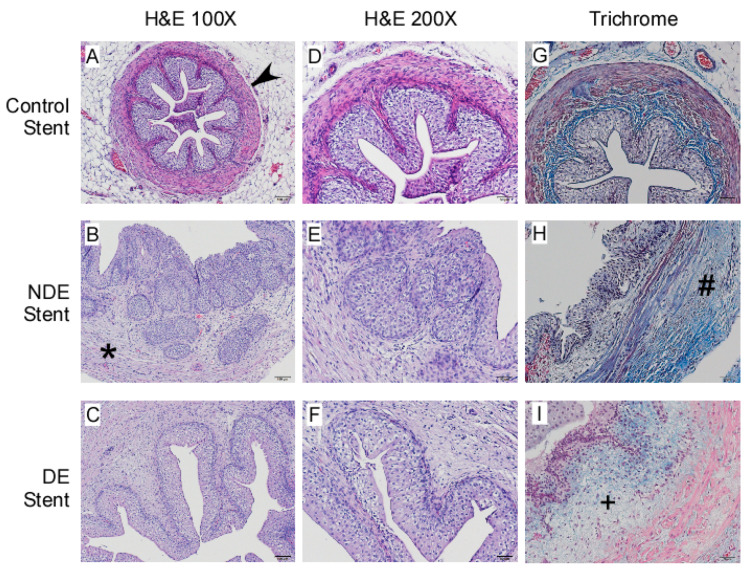
Photomicrographs showing hematoxylin and eosin (**A**–**F**), and Masson’s trichrome (**G**–**I**) staining of ureter cross-sections. (**A**). Histopathology changes of 100× magnification (**A**–**C**), 200× magnification (**D**–**F**), and Masson’s Trichrome stain at 200× magnification for collagen and smooth muscle distribution (**G**–**I**). (**A**,**B**) Circular smooth muscle decreased (arrowhead) when ureter fibrosis dominated (asterisk). (**H**,**I**) Collagen deposition was prominent in thermally injured ureters (hashtag) but was decreased in the drug-eluting group (plus).

**Figure 4 ijms-22-05664-f004:**
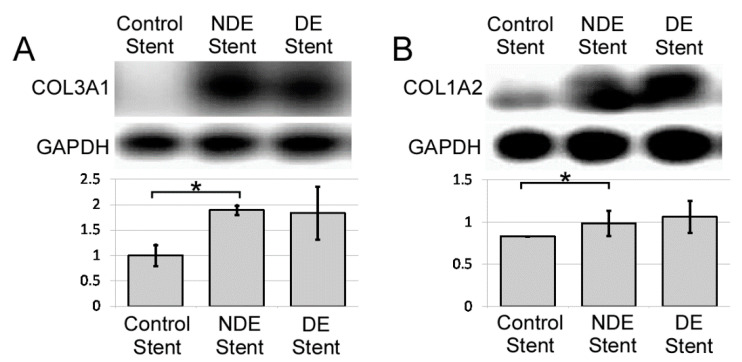
Western blot of collagen expression in ureteral protein extraction. Collagen subtypes, including (**A**). type III collagen, COL3A1, and (**B**). type I collagen, COL1A2, increased in thermally injured ureters. There was little difference in collagen expression between drug-eluting and non-eluting stents. However, the thermal injury did promote the expression of both subtypes. * *p* < 0.05 represents a significant difference for the NDE stent group compared with the control stent group.

**Figure 5 ijms-22-05664-f005:**
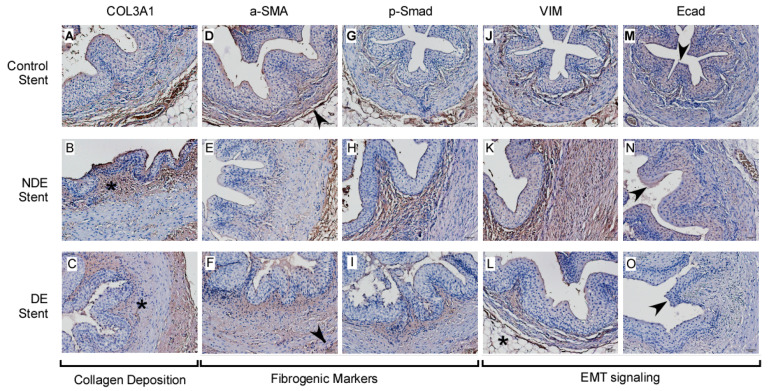
Ureter cross-section special stain (200× magnification) for expression of (**A**–**C**). collagen, (**D**–**F**). α-SMA, (**G**–**I**). p-Smad, and EMT markers including (**J**–**L**). Vimentin (VIM) and (**M**–**O**). E-cadherin (Ecad). Collagen, p-Smad, and VIM significantly increased in the thermally injured ureter (i.e.,NDE stent groups) but decreased with rapamycin treatment (i.e., DE stent groups). (**B**) Collagen COL3A1 expression increased in thermally injured ureters (asterisk). (**C**) Rapamycin eluted from DE stent result in collagen deposition decrease (asterisk). (**D**–**F**) α-SMA expression decreased significantly due to loss of smooth muscle and myofibroblast in injured ureters (comparing arrowhead). (**L**) Vimentin decreased when exposed to rapamycin (asterisk) comparing to (**K**). (**M**–**O**) E-cadherin expression is mainly localized in the urothelium (arrowheads).

**Figure 6 ijms-22-05664-f006:**
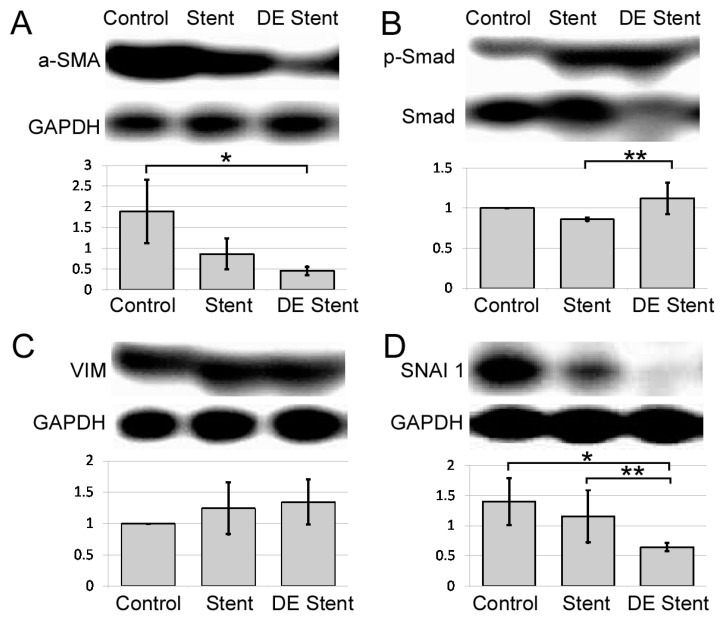
Semi-quantification of epithelial–mesenchymal transition and TGF-β signaling proteins by Western blot analysis. (**A**). α-SMA expression over GAPDH, (**B**). Phosphorylated Smad over total Smad, (**C**). Vimentin over GAPDH, (**D**). SNAI1 over GAPDH. * *p* < 0.05 represents a significant difference for the DE stent group compared with the control stent group. ** *p* < 0.05 represents a significant difference for the DE stent group compared with the NDE stent group.

## Data Availability

The data presented in this study may be available on request from the corresponding author. The data are not publicly available due to equipment dependent data.

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
