# Peer review of "Biodegradable Stent with mTOR Inhibitor-Eluting Reduces Progression of Ureteral Stricture"

_ijms, 2021, doi:10.3390/ijms22115664_

Round 1

Reviewer 1 Report

The article is interesting and the usefulness for clinical practice is intriguing.

I suggest that a different rabbit is used as control.

Author Response

REVIEW OPINONS

The article is interesting and the usefulness for clinical practice is intriguing.

I suggest that a different rabbit is used as control.

REPLY

Thank you for your delicate review and to-the-point suggestion. The current experiment design is aimed to minimize the effect of all variables except the independent variable. The independent variable in our study is the application of mTOR inhibitor, i.e. rapamycin, in the biodegradable stent. Because the control group, receiving no independent intervention, is used as a baseline to compare groups and assess the effect of that intervention, we would try to make other variable best remain the same in the control and the experiment group.

The ureters in different rabbits would have more differences, including general blood circulation, immune responses, nutrition status, etc. Comparing ureters in the same rabbit may have the best chance of keeping variables the same. On the other hand, rabbits’ mortality is high when both ureters suffered from severe fibrosis. They may only survive the healing period if dialysis nephrostomy is applied in obstructed kidney. On the other hand, when both stents (NDE & DE stents) are inserted in the same ureter, we can keep the urinary flow through both stents the same.

Reviewer 2 Report

The study by Ho et al regarding mTOR inhibition on fibrosis signaling pathways are very interesting and exciting study. I have just few points that authors may address.

  • Please change the title of the paper in a translational point that will get more attention by the reader.
  • UUO is a murine model of kidney fibrosis, in the discussion part, author may give some reference and connect the present study.
  • Did author try to measure collagen accumulation by other methods to confirm it? For example, Sircol collagen assay.
  • Is there any evidence in this study that mTOR inhibition may also affect growth of cell since mTOR signaling is involved in cellular growth?
  • Please provide full gel of all western blot.

Author Response

REVIEW OPINIONS:

The study by Ho et al regarding mTOR inhibition on fibrosis signaling pathways are very interesting and exciting study. I have just few points that authors may address.

  • Please change the title of the paper in a translational point that will get more attention by the reader.
  • UUO is a murine model of kidney fibrosis, in the discussion part, author may give some reference and connect the present study.
  • Did author try to measure collagen accumulation by other methods to confirm it? For example, Sircol collagen assay.
  • Is there any evidence in this study that mTOR inhibition may also affect growth of cell since mTOR signaling is involved in cellular growth?
  • Please provide full gel of all western blot.

REPLY

Thank you for your delicate opinions and important points.

  • Please change the title of the paper in a translational point that will get more attention by the reader.

We had changed the title to “Biodegradable Stent with mTOR inhibitor-Eluting Reduces Progression of Ureteral Stricture”. But we are also open to better suggestions.

  • UUO is a murine model of kidney fibrosis, in the discussion part, author may give some reference and connect the present study.

We had a paragraph in discussion for the link of reference to our study. “Although unilateral ureteral obstruction (UUO) is a model initially setup in mice for the study renal fibrosis, including BALB/c[1,2], C57BL/6[3,4], ICR[5,6], etc., rats had also been used for the same purpose[7]. Sprague-Dawley rats[8,9], along with Wistar rats[10,11] had been both used to study renal fibrosis in UUO. Even porcine model had been established[12,13]. Previous studies had focused on the effect of mesenchymal stem cells on rat ureteral stricture[14]. We modified the experiment but focused on using drug-eluting biodegradable stent to treat ureteral fibrotic process, which is the primary cause of hydronephrosis and renal fibrosis.

  • Did author try to measure collagen accumulation by other methods to confirm it? For example, Sircol collagen assay.

Thank you for your suggestion. Since this is the first study on rapamycin drug-eluting stent for ureteral fibrosis, we focused on the phenotype changes first. So only semi-quantitative test such as Western blot was done along with pathology evaluation. But definitely, quantitative assays will be the next step.

  • Is there any evidence in this study that mTOR inhibition may also affect growth of cell since mTOR signaling is involved in cellular growth?

Rapamycin had been well-known to inhibit vascular smooth muscle cell proliferation in stented coronary artery[15,16]. The evidences of cell growth inhibition by rapamycin is also noted in myofibroblast at other sites [17]. Though only decreased nucleus numbers, a sign of decreased cellular numbers, were found in ureter cross section staining. “Rapamycin-eluting biodegradable stents had been used in coronary artery diseases and had been shown to inhibit vascular smooth muscle cell proliferation[15,16]. This may help prevent restenosis of the vessels due to increased cells at the injury site. Besides, myofibroblast growth had also been found to be inhibited by mTOR inhibitors in kidney[18], liver[19], cornea[17] and nasal cavity[20]. Although cell growth and numbers were not quantitatively evaluated in our study, cell nuclei numbers can be found decreased in asterisk region between Fig5B and 5C along with collagen deposition. This may imply that cell proliferation was affected. Further quantitative studies are necessary to reveal the effect of rapamycin-eluting stents on cell proliferation.

  • Please provide full gel of all western blot.

The original gel photos are provided in a compressed file: Allgel-mTOR.zip

  1. Honma, S.; Shinohara, M.; Takahashi, N.; Nakamura, K.; Hamano, S.; Mitazaki, S.; Abe, S.; Yoshida, M. Effect of cyclooxygenase (cox)-2 inhibition on mouse renal interstitial fibrosis. Eur J Pharmacol 2014, 740, 578-583.
  2. Wang, M.; Chen, D.Q.; Chen, L.; Cao, G.; Zhao, H.; Liu, D.; Vaziri, N.D.; Guo, Y.; Zhao, Y.Y. Novel inhibitors of the cellular renin-angiotensin system components, poricoic acids, target smad3 phosphorylation and wnt/β-catenin pathway against renal fibrosis. British journal of pharmacology 2018, 175, 2689-2708.
  3. Baba, I.; Egi, Y.; Utsumi, H.; Kakimoto, T.; Suzuki, K. Inhibitory effects of fasudil on renal interstitial fibrosis induced by unilateral ureteral obstruction. Mol Med Rep 2015, 12, 8010-8020.
  4. Anorga, S.; Overstreet, J.M.; Falke, L.L.; Tang, J.; Goldschmeding, R.G.; Higgins, P.J.; Samarakoon, R. Deregulation of hippo-taz pathway during renal injury confers a fibrotic maladaptive phenotype. FASEB journal : official publication of the Federation of American Societies for Experimental Biology 2018, 32, 2644-2657.
  5. Tasanarong, A.; Kongkham, S.; Khositseth, S. Dual inhibiting senescence and epithelial-to-mesenchymal transition by erythropoietin preserve tubular epithelial cell regeneration and ameliorate renal fibrosis in unilateral ureteral obstruction. Biomed Res Int 2013, 2013, 308130.
  6. Xia, Z.; Abe, K.; Furusu, A.; Miyazaki, M.; Obata, Y.; Tabata, Y.; Koji, T.; Kohno, S. Suppression of renal tubulointerstitial fibrosis by small interfering rna targeting heat shock protein 47. Am J Nephrol 2008, 28, 34-46.
  7. Martínez-Klimova, E.; Aparicio-Trejo, O.E.; Tapia, E.; Pedraza-Chaverri, J. Unilateral ureteral obstruction as a model to investigate fibrosis-attenuating treatments. Biomolecules 2019, 9, 141.
  8. Xianyuan, L.; Wei, Z.; Yaqian, D.; Dan, Z.; Xueli, T.; Zhanglu, D.; Guanyi, L.; Lan, T.; Menghua, L. Anti-renal fibrosis effect of asperulosidic acid via tgf-β1/smad2/smad3 and nf-κb signaling pathways in a rat model of unilateral ureteral obstruction. Phytomedicine 2019, 53, 274-285.
  9. Wongmekiat, O.; Leelarungrayub, D.; Thamprasert, K. Alpha-lipoic acid attenuates renal injury in rats with obstructive nephropathy. Biomed Res Int 2013, 2013, 138719.
  10. Hosseinian, S.; Rad, A.K.; Bideskan, A.E.; Soukhtanloo, M.; Sadeghnia, H.; Shafei, M.N.; Motejadded, F.; Mohebbati, R.; Shahraki, S.; Beheshti, F. Thymoquinone ameliorates renal damage in unilateral ureteral obstruction in rats. Pharmacological reports : PR 2017, 69, 648-657.
  11. Hammad, F.T.; Lubbad, L. Does curcumin protect against renal dysfunction following reversible unilateral ureteric obstruction in the rat? European surgical research. Europaische chirurgische Forschung. Recherches chirurgicales europeennes 2011, 46, 188-193.
  12. Nakada, S.Y.; Soble, J.J.; Gardner, S.M.; Wolf, J.S., Jr.; Figenshau, R.S.; Pearle, M.S.; Humphrey, P.A.; Clayman, R.V. Comparison of acucise endopyelotomy and endoballoon rupture for management of secondary proximal ureteral stricture in the porcine model. J Endourol 1996, 10, 311-318.
  13. Chiu, A.W.; Lin, C.H.; Huan, S.K.; Liu, C.J.; Lin, C.C.; Huang, Y.L.; Lin, W.L.; Huang, S.H.; Lee, P.S.; Lin, C.N. Creation of ureteropelvic junction obstruction and its correction by chemical glue-assisted laparoscopic dismembered pyeloplasty. J Endourol 2003, 17, 23-28.
  14. Luo, J.; Zhao, S.; Wang, J.; Luo, L.; Li, E.; Zhu, Z.; Liu, Y.; Kang, R.; Zhao, Z. Bone marrow mesenchymal stem cells reduce ureteral stricture formation in a rat model via the paracrine effect of extracellular vesicles. J Cell Mol Med 2018, 22, 4449-4459.
  15. Rosner, D.; McCarthy, N.; Bennett, M. Rapamycin inhibits human in stent restenosis vascular smooth muscle cells independently of prb phosphorylation and p53. Cardiovascular Research 2005, 66, 601-610.
  16. Buellesfeld, L.; Grube, E. Abt-578-eluting stents. The promising successor of sirolimus- and paclitaxel-eluting stent concepts? Herz 2004, 29, 167-170.
  17. Milani, B.Y.; Milani, F.Y.; Park, D.W.; Namavari, A.; Shah, J.; Amirjamshidi, H.; Ying, H.; Djalilian, A.R. Rapamycin inhibits the production of myofibroblasts and reduces corneal scarring after photorefractive keratectomy. Invest Ophthalmol Vis Sci 2013, 54, 7424-7430.
  18. Kern, G.; Mair, S.M.; Noppert, S.J.; Jennings, P.; Schramek, H.; Rudnicki, M.; Mueller, G.A.; Mayer, G.; Koppelstaetter, C. Tacrolimus increases nox4 expression in human renal fibroblasts and induces fibrosis-related genes by aberrant tgf-beta receptor signalling. PLoS One 2014, 9, e96377.
  19. Masola, V.; Carraro, A.; Zaza, G.; Bellin, G.; Montin, U.; Violi, P.; Lupo, A.; Tedeschi, U. Epithelial to mesenchymal transition in the liver field: The double face of everolimus in vitro. BMC Gastroenterol 2015, 15, 118.
  20. Ko, D.Y.; Shin, J.M.; Um, J.Y.; Kang, B.; Park, I.H.; Lee, H.M. Rapamycin inhibits transforming growth factor beta 1 induced myofibroblast differentiation via the phosphorylated-phosphatidylinositol 3-kinase mammalian target of rapamycin signal pathways in nasal polyp-derived fibroblasts. American journal of rhinology & allergy 2016, 30, 211-217.

Reviewer 3 Report

This is an interesting study with limited clinical significance. Ureteral injuries caused and recognized at surgery are readily repaired with stents. If more extensive with ureteral/bladder/kidney mobilization and repair. The clinical scenario to test is if after injury noted post-surgery at various time points - can drug eluting stents help maintain patency and repair from endoscopic incision or dilation

Author Response

Review opinions

This is an interesting study with limited clinical significance. Ureteral injuries caused and recognized at surgery are readily repaired with stents. If more extensive with ureteral/bladder/kidney mobilization and repair. The clinical scenario to test is if after injury noted post-surgery at various time points - can drug eluting stents help maintain patency and repair from endoscopic incision or dilation.

REPLY

Thank you for the opinions. Ureter injuries can definitely be repair by experienced urological surgeons[1]. And all kinds of ureter stricture and stenosis can be managed with pyeloplasty[2], ureteroureterostomy/  transureteroureterostomy [3,4], or psoas hitch/ Boari flap[5] in upper, middle, and lower ureter injuries respectively. But fibrosis is always a challenge for surgeons because this abnormal healing process may ruin a great operation. We hope to reduce surgical site injury-induced ureteral fibrosis to make perfect surgeries better. If rapamycin eluting biodegradable stent can work efficiently at stented operation site, fibrosis may be reduced with promised ureteral patency. We had added descriptions in the article. “Ureter injuries can well managed by experienced urological surgeons[1]. And different kinds of ureter stricture and stenosis can be managed with pyeloplasty[2], ureteroureterostomy/  transureteroureterostomy [3,4], or psoas hitch/ Boari flap[5] in upper, middle, or lower ureter injuries respectively. But fibrosis is always a challenge for surgeons because this abnormal healing process may ruin a great operation. Besides,

References:

  1. Vasudevan, V.P.; Johnson, E.U.; Wong, K.; Iskander, M.; Javed, S.; Gupta, N.; McCabe, J.E.; Kavoussi, L. Contemporary management of ureteral strictures. Journal of Clinical Urology 2019, 12, 20-31.
  2. Wang, Q.; Lu, Y.; Hu, H.; Zhang, J.; Qin, B.; Zhu, J.; Dirie, N.I.; Zhang, Z.; Wang, S. Management of recurrent ureteral stricture: A retrospectively comparative study with robot-assisted laparoscopic surgery versus open approach. PeerJ 2019, 7, e8166-e8166.
  3. Derrick, F.C., Jr.; Lynch, K.M., Jr.; Price, R., Jr.; Turner, W.R., Jr. Transureteroureterostomy. JAMA 1967, 200, 987-990.
  4. Andrade, H.S.; Kaouk, J.H.; Zargar, H.; Caputo, P.A.; Akca, O.; Ramirez, D.; Autorino, R.; Noble, M.; Stein, R.J. Robotic ureteroureterostomy for treatment of a proximal ureteric stricture. International braz j urol : official journal of the Brazilian Society of Urology 2016, 42, 1041-1042.
  5. Stein, R.; Rubenwolf, P.; Ziesel, C.; Kamal, M.M.; Thüroff, J.W. Psoas hitch and boari flap ureteroneocystostomy. BJU International 2013, 112, 137-155.

Round 2

Reviewer 1 Report

I am completed satisfied of the AA's response